# Evaluation of therapeutic agent selection based on comprehensive genomic profiling in gastroenteropancreatic neuroendocrine neoplasms

**Suguru Miyazawa[1], Hiroaki Ono[1]\*, Hironari Yamashita[1], Daisuke Asano[1], Yoshiya Ishikawa[1], Shuichi Watanabe[1], Hiroki Ueda[1], Satoru Aoyama[2], Naoya Ishibashi[2], Keiichi Akahoshi[1], Sadakatsu Ikeda[2], Minoru Tanabe[1]**

**1** Department of Hepatobiliary and Pancreatic Surgery, Graduate School of Medicine, Institute of Science Tokyo, Tokyo, Japan, **2** Department of Precision Cancer Medicine, Center for Innovative Cancer Treatment, Institute of Science Tokyo, Tokyo, Japan

\* ono.msrg@tmd.ac.jp

## Abstract

### Introduction

Comprehensive genomic profiling (CGP) is increasingly being integrated into standard clinical practice as a strategy to guide subsequent treatment decisions by identifying novel therapeutic options based on tumor-specific mutations. However, its clinical utility in neuroendocrine neoplasms (NENs) remains to be determined. We conducted a cross-sectional analysis of genomic alterations, including tumor mutational burden (TMB) and microsatellite instability (MSI), in gastroenteropancreatic neuroendocrine neoplasms (GEP-NENs), comparing neuroendocrine tumors (NETs) and neuroendocrine carcinomas (NECs).

### Results

CGP was performed on 50 patients with advanced GEP-NENs between August 2017 and October 2023. Of these, 38 were diagnosed with NETs and 12 with NECs. Primary tumor sites included the pancreas (n = 24), gastrointestinal tract (n = 25), and one case of unknown origin. In our study, CGP analysis comparing NETs and NECs documented different genetic profiles. In addition, median TMB was significantly higher in NECs (5.0 vs. 1.9 mutations/megabase; $p = 0.0101$). High TMB was identified in 4 of 12 NECs (33.3%) and in only 1 of 38 NETs (2.6%) ($p = 0.0093$). NECs also harbored a significantly greater number of mutations per case than NETs (5.5 vs. 2.0; $p = 0.0014$). Actionable mutations were identified in 18 of 50 patients, and 7 patients received mutation-guided therapy: 2 with NETs and 5 with NECs. The frequency of treatment initiation based on CGP findings was significantly higher in patients with NECs ($p = 0.0059$).

**Data availability statement:** All relevant data are within the paper and its Supporting Information files.

**Funding:** The author(s) received no specific funding for this work.

**Competing interests:** The authors have declared that no competing interests exist.

## Conclusions

NECs are genetically distinct from NETs, with a higher prevalence of actionable mutations and greater therapeutic relevance of CGP findings. These results suggest that CGP may be particularly valuable in NECs, where therapeutic options are limited, supporting its proactive implementation in this subgroup.

## Introduction

Neuroendocrine neoplasms (NENs) comprise a heterogeneous group of tumors originating from neuroendocrine cells; they exhibit characteristics of both neuronal and hormone-secreting endocrine cells. NENs can develop in various organs, with approximately 60% arising in the gastrointestinal tract or pancreas (gastroenteropancreatic neuroendocrine neoplasms, GEP-NENs) and around 20–30% occurring in the lungs [1,2]. Recent studies indicate a significant global increase in both the prevalence and incidence of GEP-NENs, primarily attributed to advances in endoscopic and imaging diagnostic techniques [1–4].

NENs are broadly classified into neuroendocrine tumors (NETs) and neuroendocrine carcinomas (NECs) based on their histopathological characteristics and biological behavior [5]. NETs are well-differentiated tumors with relatively slow progression and are further graded into G1, G2, and G3 according to the Ki-67 proliferation index and mitotic count. In contrast, NECs are poorly differentiated, highly aggressive carcinomas with a high proliferation index. Given their distinct biological behaviors, NETs and NECs require different therapeutic approaches.

At diagnosis, 20–30% of GEP-NENs present with metastases [6]. The liver is the most common site of metastasis, involved in approximately 82% of metastatic GEP-NEN cases [7]. Patients with liver metastases have a significantly lower five-year survival rate than those without metastases [8]. For these cases, it is essential to combine systemic therapies—such as molecular-targeted drugs, chemotherapy, and peptide receptor radionuclide therapy (PRRT)—with local therapies, including surgery, transarterial chemoembolization (TACE), and ablation [9,10].

Comprehensive genomic profiling (CGP) is an advanced molecular diagnostic approach to analyze cancer-related genetic alterations in tumor tissue or blood samples. Utilizing next-generation sequencing (NGS), CGP simultaneously assesses a broad spectrum of genomic abnormalities, including single nucleotide variants (SNVs), insertions and deletions (indels), copy number alterations (CNAs), gene fusions, and structural rearrangements. Recently, in Japan, comprehensive genomic profiling (CGP) has been approved for coverage under national health insurance for patients expected to complete standard therapy. It has become a standard approach in oncology, playing a crucial role in precision medicine by guiding the selection of targeted therapies and informing treatment strategies.

Pembrolizumab, an immune checkpoint inhibitor, is approved for multiple cancer types, particularly those with high tumor mutational burden (TMB-high) or high microsatellite instability (MSI-high) [11,12]. To date, several reports have evaluated

the effectiveness of pembrolizumab in treating NENs, with a response rate of only 3.4–6.7% [13–15]. In all cases, PD-L1 expression was used as a biomarker. Meanwhile, isolated case reports have documented tumor shrinkage following pembrolizumab administration in TMB-high or MSI-high cases [16,17].

Previous studies, though limited, suggest that cases of GEP-NENs with high TMB or MSI are rare, with reported frequencies ranging from 1% to 6% [16,18,19]. However, in NECs, the frequency may be higher; one study identified high TMB in 4 out of 16 cases [20]. To date, no studies have cross-sectionally analyzed TMB and MSI across different tumor grades. Furthermore, data on treatment outcomes following genomic analysis remain limited, particularly regarding the frequency with which such results lead to actual drug administration and the subsequent treatment course.

Therefore, this study aimed to perform a detailed analysis of genetic mutations, TMB, and MSI status in advanced GEP-NENs, with a particular focus on comparing NETs and NECs based on genetic panel testing results. Additionally, this study aimed to assess the frequency with which the panel test results influence actual treatment decisions and to evaluate the corresponding treatment outcomes, which are crucial for developing effective treatment strategies for GEP-NENs. Furthermore, we present two cases of pembrolizumab treatment in patients with GEP-NENs, both of whom exhibited excellent clinical responses, suggesting that CGP may be more beneficial in NECs.

## Methods

### Patients

Fifty patients with advanced gastroenteropancreatic neuroendocrine neoplasms (GEP-NENs) who underwent genomic sequencing at Tokyo Medical and Dental University Hospital were enrolled between August 1, 2017, and October 31, 2023. The data were accessed for research purposes on November 27, 2023.

This study was conducted with the approval of the Ethics Committee of the Tokyo Medical and Dental University Faculty of Medicine (approval number M2000-1080, G2017-018–07), and written informed consent for the use of medical record data for research purposes was obtained from all patients. The patients were anonymized and coded in accordance with the ethical guidelines stipulated in the Declaration of Helsinki. As of October 2024, the university and hospital were renamed Institute of Science Tokyo and Institute of Science Tokyo Hospital, respectively.

### Complete genome profiling (CGP)

Comprehensive genome profiling (CGP) is a high-throughput molecular technique employed to detect clinically relevant genetic alterations in tumor specimens or circulating cell-free DNA. For this analysis, tissue samples were obtained via surgical resection or biopsy and subsequently formalin-fixed and paraffin-embedded (FFPE) as pathological sections. When sufficient tissue samples were not available or when tissue analysis did not yield adequate results, comprehensive genomic profiling (CGP) was performed using blood samples.

In our study, we investigated mutations based on comprehensive genomic profiling using two commercially available assays. Primarily, FoundationOne® CDx was utilized for tissue samples; this is a next-generation sequencing test that evaluates the complete coding sequences of 324 cancer-related genes. For liquid biopsy samples, Guardant360® CDx, a comprehensive genomic profiling test capable of detecting single nucleotide variations (SNVs), indels, fusions, and copy number alterations across 74 cancer-related genes, was used. The methodology for this type of CGP has been described previously [21–24].

Based on the results of CGP, clinically significant genomic alterations were identified and assessed for their potential therapeutic relevance. In this context, particular attention was given to the presence of actionable mutations. An actionable mutation is defined as a genomic alteration that satisfies the following criteria: the change in the gene is characterized as a target for molecular targeted therapy, and a drug is available for human use, either as an antibody or as a small molecule with an inhibitory concentration of 50% (IC50) in the nanomolar range [25,26]. These mutations are considered to have potential clinical utility, guiding treatment strategies and enabling personalized cancer therapy.

## Molecular tumor board

Following the results of the genomic tests, each case was discussed by a Molecular Tumor Board (MTB) composed of specialists including medical oncologists, pathologists, radiologists, bioinformaticians, genetic counselors, clinical research coordinators, and treating physicians. These experts discussed effective treatment strategies based on CGP results, taking into consideration existing evidence and literature. They also reviewed clinical trials and investigational studies suitable for patient participation based on genomic findings, as well as the potential off-label use of existing therapeutic agents, and provided this information to the patients.

## Statistical analysis

Continuous variables were compared between two groups using the Wilcoxon test, and for comparisons among three or more groups, the Kruskal-Wallis test was utilized. For qualitative variables, Fisher's exact test or the chi-squared test was employed. All P-values were derived from two-tailed tests, with statistical significance set at $P < 0.05$.

## Results

### Clinical features

Based on the WHO 2019 classification, there were 5 cases of NET-G1, 26 cases of NET-G2, 7 cases of NET-G3, and 12 cases of NEC. Of these, 24 cases were pancreatic NENs (P-NENs), 25 were gastrointestinal NENs, and 1 had an unknown primary origin (Table 1). Gastrointestinal NENs were located in the rectum (n = 12), duodenum (n = 4), small intestine (n = 3), stomach (n = 2), bile duct (n = 2), cecum (n = 1), and gallbladder (n = 1). The median number of metastatic sites among the 50 GEP-NEN cases was 2. Liver metastases were present in 94% of patients, lymph node metastases in 70%, and lung and bone metastases in 22%.

Of the 50 cases analyzed, 38 were NETs and 12 were NECs (Table 2). Among the NETs, pancreatic NETs were the most common (20 cases, 52.6%), followed by rectal NETs (10 cases, 26.3%). Other primary sites included the small intestine, duodenum, stomach, and bile duct. There was one case each of multiple endocrine neoplasia type 1 (MEN1) and Von Hippel–Lindau (VHL) syndrome, as well as one functional NET (insulinoma). Among the NECs, pancreatic NECs were the most frequent (4 of 12 cases, 33.3%), followed by duodenal and rectal NECs, with two cases each (16.7%).

Comparison of clinical features revealed that both NETs and NECs frequently presented with liver and lymph node metastases; however, liver metastases were significantly more common in NETs. In contrast, skin metastases were significantly more frequent in NECs (Table 2).

### The frequency of gene mutations in GEP-NENs according to CGP tests

In the present study, among the 50 patients with advanced GEP-NENs who underwent CGP tests, 33 cases (66%) were analyzed using FoundationOne® CDx alone, 13 cases (26%) using Guardant360® CDx alone, and 4 cases (8%) using both methods.

Mutations were identified in 44 cases (88%), of which 18 (36%) were actionable and 7 patients (14%) were subsequently treated (Fig 1). Of the 32 NETs with identified gene mutations, 10 were actionable, two were actually treated. In contrast, mutations were identified in all NECs, with 8 being actionable (66.7%) and 5 then treated (41.7%). Thus, the frequency of mutations leading to treatment initiation was significantly higher in NECs ($p = 0.0059$)

### Mutation profiles of NETs and NECs

Differences in mutation profiles between NETs and NECs were evaluated (Figs 2 and 3). *TP53* mutations were the most frequently observed in both NETs and NECs, occurring in 8 of 38 NETs (21.1%) and 7 of 12 NECs (58.3%). Mutations

**Table 1. Baseline characteristics of 50 patients with NENs.**

| Characteristics | Total n = 50 |
|---|---|
| Age, median (range) | 58 (21-84) |
| Sex, male/female | 27/23 |
| Genetic syndrome, n (%) | 2 (4.0) |
| MEN type I | 1 (2.0) |
| VHL | 1 (2.0) |
| Functionality, nonfunctioning, n (%) | 1 (2.0) |
| Ki-67 index, median (range) | 14.3 (0.2-90) |
| Tumor Grade, n (%) | |
| NET G1 | 5 (10.0) |
| NET G2 | 26 (52.0) |
| NET G3 | 7 (14.0) |
| NEC G3 | 12 (24.0) |
| Primary tumor site, n (%) | |
| Pancreas | 24 (48.0) |
| Rectum | 12 (24.0) |
| Duodenum | 4 (8.0) |
| Ileum | 3 (6.0) |
| Bile duct | 2 (4.0) |
| Stomach | 2 (4.0) |
| Cecum | 1 (2.0) |
| Gallbladder | 1 (2.0) |
| Unknown | 1 (2.0) |
| Number of metastases, median (range) | 2 (1-5) |
| Site of metastases, n (%) | |
| Liver | 47 (94.0) |
| Lymph node | 35 (70.0) |
| Lung | 11 (22.0) |
| Bone | 11 (22.0) |
| Peritoneum | 4 (8.0) |
| Adrenal gland | 3 (6.0) |
| Skin | 3 (6.0) |
| Brain | 2 (4.0) |
| Pancreas | 2 (4.0) |
| Pleura | 1 (2.0) |

Note: Continuous variables are expressed as medians and ranges. Qualitative variables are expressed as numbers (%).

Abbreviations: MEN, multiple endocrine neoplasia: VHL, Von Hippel-Lindau syndrome.

characteristic of NETs included *MEN1* (6 cases, 15.8%) and *DAXX* (5 cases, 13.2%), which were not detected in NECs. In contrast, *RB1* and *KRAS* mutations were exclusively observed in NECs, detected in 6 cases (50%) and 3 cases (25.0%), respectively. These mutations were rare in NETs, with *RB1* found in only one case. Consistent with previous reports, NETs and NECs displayed divergent genetic profiles [27].

The figure illustrates tumor mutational burden (TMB-high) and individual gene mutations in NETs classified by grades G1–G3. Mutation frequencies are shown as the number and percentage of cases.

**Table 2. Baseline characteristics of patients comparing with NETs or NECs.**

| Characteristics | NETs (n = 38) | NECs (n = 12) | P value |
|---|---|---|---|
| Age, median(range) | 56 (21-79) | 69 (48-84) | 0.0345 |
| Sex, male/female | 21/17 | 6/6 | 0.99 |
| Genetic syndrome, n (%) | 2 (5.3) | 0 | 0.99 |
| MEN type I | 1 (2.6) | 0 | 0.24 |
| VHL | 1 (2.6) | 0 | 0.99 |
| Functionality, nonfunctioning | 1 (2.6) | 0 | 0.99 |
| Ki-67 index, mean ±SD | 13.1±10.0 | 65.1±15.5 | <0.0001 |
| Primary tumor site, n (%) | | | |
| Pancreas | 20 (52.6) | 4 (33.3) | 0.327 |
| Rectum | 10 (26.3) | 2 (16.7) | 0.705 |
| Ileum | 3 (7.9) | 0 | 0.99 |
| Duodenum | 2 (5.3) | 2 (16.7) | 0.24 |
| Stomach | 2 (5.3) | 0 | 0.99 |
| Bile duct | 1 (2.6) | 1 (8.3) | 0.426 |
| Gallbladder | 0 | 1 (8.3) | 0.24 |
| Cecum | 0 | 1 (8.3) | 0.24 |
| Unknown | 0 | 1 (8.3) | 0.24 |
| Number of metastases, median(range) | 2 (1-5) | 3(1-4) | 0.187 |
| Site of metastases, n (%) | | | |
| Liver | 38 (100) | 9 (75.0) | 0.0112 |
| Lymph node | 25 (65.8) | 10 (83.3) | 0.304 |
| Lung | 6 (15.8) | 5 (41.7) | 0.105 |
| Bone | 7 (18.4) | 4 (33.3) | 0.424 |
| Peritoneum | 4 (10.5) | 0 | 0.56 |
| Adrenal gland | 3 (7.9) | 0 | 0.99 |
| Skin | 0 | 3 (25.0) | 0.00112 |
| Brain | 1 (2.6) | 1 (8.3) | 0.426 |
| Pancreas | 2 (5.3) | 0 | 0.99 |
| Pleura | 0 | 1 (8.3) | 0.24 |

Note: Continuous variables are expressed as medians and ranges. Qualitative variables are expressed as numbers (%).

Abbreviations: MEN, multiple endocrine neoplasia: VHL, Von Hippel-Lindau syndrome.

## The prevalence of TMB-high and gene mutations in GEP-NENs

TMB values and mutation counts in GEP-NENs are shown in Fig 4A and 4B. The median TMB was 3 mutations per megabase (range: 0–48). Median TMB values differed significantly between NETs and NECs (1.9 vs. 5.0, respectively; p = 0.0101) (Fig 4A). A high TMB was defined as more than 10 mutations per megabase, based on previous reports [7]. The incidence of high TMB was significantly higher in NECs (33.3%, 4 of 12 cases) than in NETs (2.6%, 1 of 38 cases) (p = 0.0093).

Genomic sequencing identified mutations in 44 cases (88%), with a median of 3 mutations per case (range: 0–22). The number of mutations was significantly greater in NECs than in NETs (median: 5.5 vs. 2.0, respectively; p = 0.0014) (Fig 3B).

No significant differences in TMB values were observed among the NET subgroups; the median TMB values for NET G1, G2, and G3 were 2.94, 2.45, and 1.0, respectively (p = 0.989). In contrast, a significant increase in mutation counts was associated with higher tumor grade, with median mutation counts of 1.0, 2.5, and 3.0 for NET G1, G2, and G3, respectively (p = 0.0402) (S1 Fig).

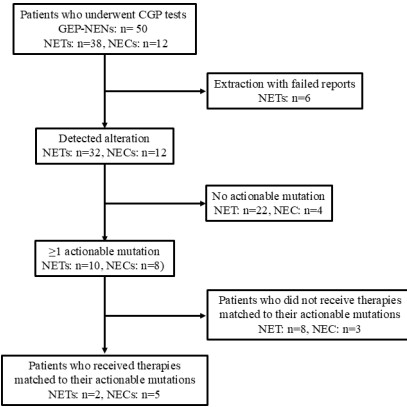

**Fig 1. Flowchart of GEP-NENs patients who underwent CGP tests.** Among 50 patients with gastroenteropancreatic neuroendocrine neoplasms (NETs: n = 38; NECs: n = 12), 44 patients had detectable genomic alterations. Actionable mutations were identified in 18 cases (NETs: n = 10; NECs: n = 8), of which 7 patients received matched targeted therapies (NETs: n = 2; NECs: n = 5).

## Actionable mutations and the corresponding therapy

Table 3 summarizes actionable mutations identified through CGP analysis, the corresponding therapeutic agents derived from these findings, and their observed clinical effects.

CGP identified actionable mutations in 10 NETs and 8 NECs. Of the 10 NET cases with actionable alterations, targeted therapy could only be initiated in two; in both of these, treatment duration was short due to poor performance status, making it difficult to assess the clinical efficacy of the administered agents. In contrast, of the 8 NEC patients with actionable mutations, 5 underwent treatments based on the identified genomic alterations. Of the latter, two experienced a partial response (PR). One patient had progressive disease (PD), while the remaining two suffered a rapid deterioration in their general condition shortly after treatment initiation, making it difficult to assess treatment efficacy. The cases with favorable responses were both TMB-high and treated with pembrolizumab, as described in the following section and illustrated in Fig 5.

The clinical courses of 12 NEC cases with distant metastases who underwent comprehensive genomic profiling (CGP) are shown in S2A Fig (primary tumor site, TMB, MSI status, and OS for all 12 NEC cases). Overall survival (OS) was 693 days (S2B Fig). Based on CGP results, selected therapies were administered to 5 of the 12 patients. Patients who received CGP-guided therapy demonstrated a trend toward improved prognosis compared to those who did not. (S2C Fig).

## Clinical responses to pembrolizumab for patients with TMB-high

**Patient A (Fig 5A.).**  A 67-year-old male was diagnosed with rectal NEC G3 (Ki-67 index: 70) following a low anterior resection. No synchronous distant metastases were evident at the time of diagnosis; therefore, curative resection of the rectal lesion was performed.

Seven months after surgery, brain metastases were detected, and whole-brain irradiation was administered. At nine months post-operation, subcutaneous, radial, and splenic metastases were identified, prompting the initiation of CDDP/VP16 chemotherapy. Although the disease was classified as progressive (PD), comprehensive genomic profiling (CGP) revealed both MSI-high and TMB-high (TMB = 48), leading to the initiation of pembrolizumab treatment.

Rapid tumor shrinkage was observed following pembrolizumab administration, and the patient is currently alive 60.7 months after diagnosis, maintaining a partial response (PR) for over four years.

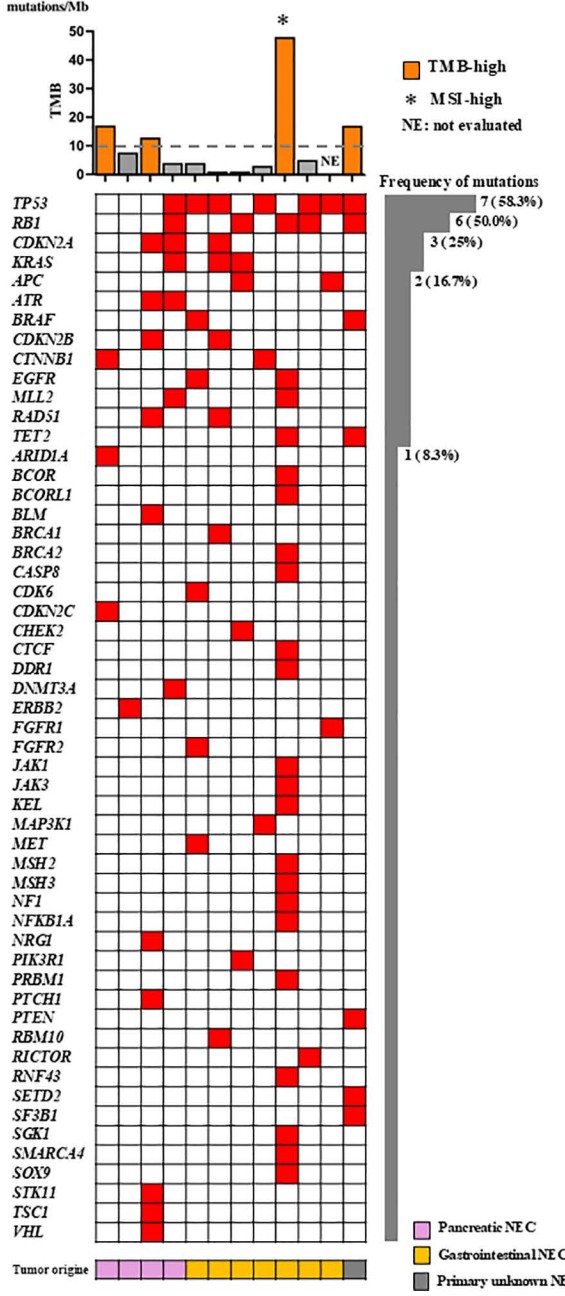

**Fig 2. Distribution of TMB-high and gene mutations in NECs.** Tumor mutational burden (TMB-high) and individual gene mutations are presented. Mutation frequencies are shown as the number and percentage of cases. "NE" indicates not evaluable.

**Patient B (Fig 5B.).** An 81-year-old male was diagnosed with NEC following a biopsy of a subcutaneous nodule on the back (Ki-67 index:70). In addition to the lesion on the right back, metastases were identified in the axilla, perisplenic region, and mesenteric lymph nodes. The primary tumor site was unclear. After resection of the large subcutaneous tumor, treatment with CDDP plus VP16 was initiated for NEC of unknown primary origin but was discontinued due to Grade 4 leukopenia as an adverse event.

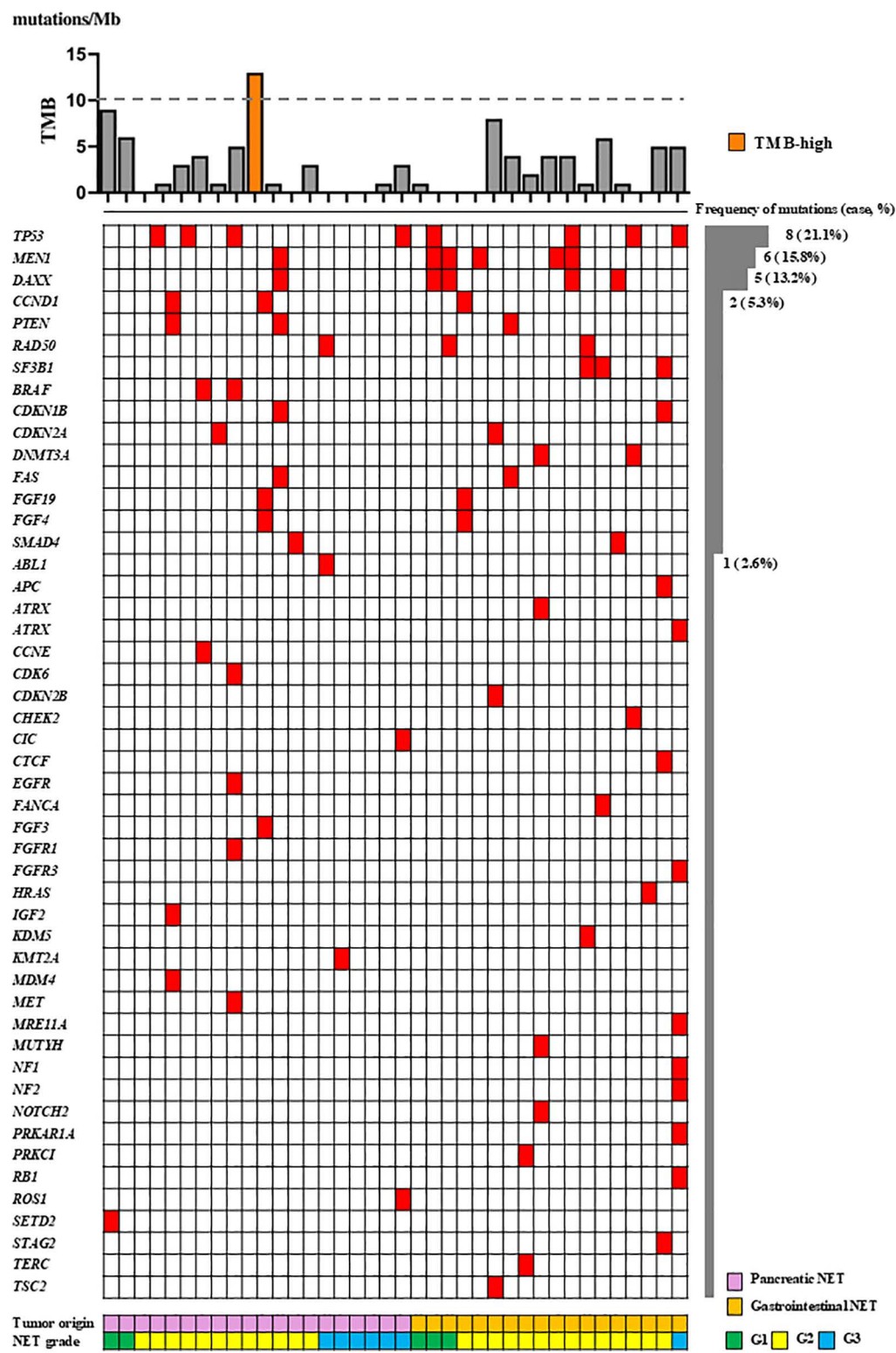

**Fig 3. Distribution of TMB-high and gene mutations in NETs.**

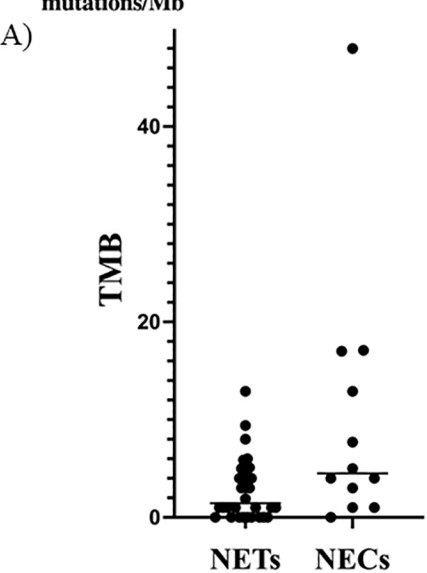

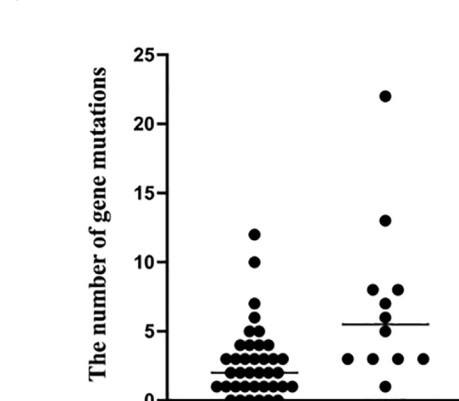

**Fig 4. Comparison of tumor mutational burden (TMB) (A) and the number of gene mutations (B) between NETs and NECs.** (A) NECs exhibited a significantly higher median TMB than NETs (5.0 vs. 1.9, respectively; $p = 0.0101$). TMB-high was defined as >10 mutations per megabase. (B) The total number of mutations identified through genomic sequencing is shown. NECs had significantly more mutations than NETs (median: 5.5 vs. 2.0, respectively; $p = 0.0014$).

Comprehensive genomic profiling (CGP) of the resected specimen revealed a TMB-high status (TMB = 17), leading to the initiation of pembrolizumab treatment. Tumor shrinkage was observed shortly thereafter; however, therapy was temporarily suspended after two courses due to interstitial pneumonia, an immune-related adverse event (irAE). The condition subsequently improved.

The patient remains alive 9 months after diagnosis, maintaining a partial response (PR) for 5 months despite not resuming pembrolizumab therapy.

## Discussion

To the best of our knowledge, this is the first report to analyze genetic panel testing in gastroenteropancreatic neuroendocrine neoplasms (GEP-NENs) with distant metastases, evaluating not only the genetic profiles but also the

**Table 3. Cases in which comprehensive genomic profiling (CGP) identified actionable mutations.**

|     | #  | Primary Site | TMB  | MSI status | Actionable mutation | Treatment | Response |
|-----|----|--------------|------|------------|---------------------|-----------|----------|
| NET | 1  | Pancreas | 12.9 | MSS | CCND1 | CDK4/6 inhibitor<br>PARP inhibitor | – |
|     | 2  | Pancreas | 9.4  | MSS | RAD50 | PARP inhibitor (Talazopalib) | NE |
|     | 3  | Duodenum | 0    | MSS | BRAF | MEK inhibitor | – |
|     | 4  | Pancreas | 0    | MSS | CDKN2A | CDK4/6 inhibitor (Palbociclib) | NE |
|     | 5  | Rectum | 1    | MSS | ABL1 | Tyrosine kinase inhibitor | – |
|     | 6  | Pancreas | 0    | MSS | MEN1 | CDK4/6 inhibitor<br>IGF-1/2 inhibitor | – |
|     | 7  | Pancreas | 5    | MSS | CDKN2A/B | CDK4/6 inhibitor | – |
|     | 8  | Gastric | 0    | MSS | PTEN | PI3K inhibitor<br>AKT inhibitor | – |
|     | 9  | Gastric | 4    | MSS | DNMT3A | DNMT inhibitor | – |
|     | 10 | Pancreas | 5    | MSS | FGFR3 | FGFR inhibitor | – |
| NEC | 1  | Pancreas | 17.1 | MSS | TMB-high | ICI (Pembrolizumab) | NE |
|     | 2  | Pancreas | 7.7  | MSS | ERBB2 | HER2 inhibitor | – |
|     | 3  | Pancreas | 12.9 | MSS | VHL | Tyrosine kinase inhibitor (Sunitinib) | PD |
|     | 4  | Duodenum | 1    | MSS | BRCA1 | PARP inhibitor (Olaparib) | NE |
|     | 5  | Duodenum | 3    | MSS | CTNB1 | β catenin inhibitor | – |
|     | 6  | Rectum | 48   | MSI | TMB-high | ICI (Pembrolizumab) | PR |
|     | 7  | Pancreas | 4    | MSS | KRAS | MEK inhibitor<br>RAS inhibitor | – |
|     | 8  | Unknown | 17   | MSS | TMB-high | ICI (Pembrolizumab) | PR |

For patients who received therapies matched to actionable mutations, the drug name and the response are presented.

For patients who could not receive therapies matched to actionable mutations, candidate therapeutic agents are presented.

In cases where the treatment period was too short to allow for a reliable assessment, "NE" (not evaluable) is indicated.

ICI: immune check point inhibitor.

frequency with which such profiling led to treatment. We found that neuroendocrine carcinomas (NECs) exhibited a higher tumor mutation burden (TMB), i.e., a greater number of mutations, compared to neuroendocrine tumors (NETs). Consequently, actionable mutations were more frequently identified in NECs, and a significantly higher proportion of patients received the recommended targeted therapy based on these findings. Previous reports on TMB values in NECs by the groups of Chalmers, Xing, and van Riet J indicated median TMB values ranging from 2.7 to 5.68 [20,28,29]. In our NEC cohort, the median TMB was 5, which is consistent with these previous findings (Fig 4). Regarding the proportion of NECs classified as TMB-high, Chalmers et al. reported 6.1% (674 cases, with TMB > 20 defined as high), while van Riet J et al. reported 25.0% (4 out of 16 cases). In our study, a significantly higher proportion of NECs were TMB-high (33.3%), which is more comparable to the latter report that focused on distant metastatic or locally advanced NENs [20].

There are few reports on TMB in NETs; Puccini and colleagues found that there were significantly more TMB-high class among high-grade (HG) GEP-NENs compared to low-grade (LG), As the HG group included both NET-G3 and NEC, a direct comparison of TMB values between NET and NEC is not possible [19]. Van Riet J et al. employed whole-genome sequencing and concluded that NECs exhibit genomic instability with diploid to triploid genomes, leading to a higher TMB of 5.45/Mb, whereas NETs are characterized by relatively stable diploid tumor genomes with limited chromosomal arm anomalies, resulting in a lower TMB of 1.09/Mb [29]. In fact, NETs are known to have lower TMB values compared to

A)

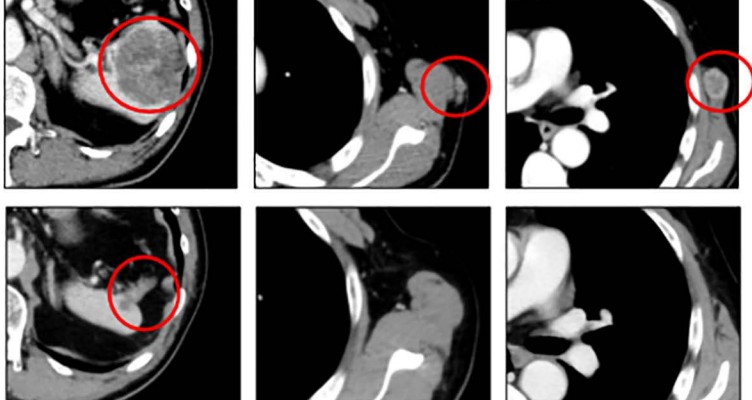

B)

**Fig 5. Representative CT images of two NEC patients treated with pembrolizumab.** (A) A patient who achieved a partial response (PR) maintained for over 4 years. (B) A patient showing PR after two courses of pembrolizumab. Baseline imaging before treatment is shown for each case.

many other cancers [30]. Our study found no significant difference in TMB levels across NET G1-G3 grades (S1A Fig), but the number of mutations significantly increased with grade (S1B Fig).

Our study indicates that 33.3% of NECs are TMB-high and therefore the treatment with pembrolizumab is possible. We have indeed treated two NEC patients with a high TMB, for whom pembrolizumab showed significant efficacy. In contrast, in NETs, even cases with distant metastases had a low frequency of TMB-high or MSI-high, leading to an anticipated lower rate of actual drug administration based on CGP tests.

According to Chalmers et al., among a large cohort of 62,150 patients with various cancer types, only 4,326 (6.96%) were classified as TMB-high, and just 1.2% exhibited both MSI-high and TMB-high status. In most cancer types, TMB-high

status is associated with microsatellite stability, with only a small subset displaying both TMB-high and MSI-high characteristics [29]. However, gastrointestinal cancers—including gastric, duodenal, small intestinal, and colorectal adenocarcinomas—demonstrate a high concordance between TMB-high and MSI-high profiles. Although data on MSI frequency in GEP-NENs remain limited, previous reports suggest that approximately 10% of gastric and colorectal NECs are MSI-high [31,32].

We report notable clinical efficacy of pembrolizumab in two cases of neuroendocrine carcinoma (NEC): one sporadic rectal NEC characterized by both TMB-high and MSI-high status, and one NEC of unknown primary origin with TMB-high but microsatellite-stable (MSS) status. While PD-L1expression is widely considered a predictive biomarker for response to PD-1/PD-L1 inhibitors in several solid tumors—including non-small cell lung cancer, urothelial carcinoma, and head and neck squamous cell carcinoma—its relevance in neuroendocrine neoplasms (NENs) remains uncertain [33–35]. Three previous studies assessing pembrolizumab in NENs reported modest efficacy rates of 3.4–6.7%, all using PD-L1 expression as the biomarker [13–15]. According to Xing et al., PD-L1-positive NECs demonstrated significantly worse overall survival than PD-L1-negative NECs. In contrast, our study observed prolonged overall survival in NEC patients treated with pembrolizumab (S2A Fig), suggesting that PD-L1 expression may not be a reliable predictor of response in NEC. Instead, TMB-high and MSI-high status appear to have greater clinical relevance.

Among the 12 NEC cases analyzed, several harbored actionable mutations detected by CGP. However, in multiple instances, the patient's clinical condition had deteriorated by the time CGP results were available, precluding the initiation of personalized therapy. The frequency of actionable alterations was markedly higher in NECs than in NETs, highlighting the importance of early CGP implementation upon diagnosis of NEC.

Molecular profiling also supports the notion that NECs are genetically distinct from NETs, consistent with previous reports [27]. Further elucidation of the tumorigenic mechanisms underlying NEC may facilitate the development of more effective treatment strategies.

Limitations of this study include its retrospective nature, the single-institution setting, and the small sample size, which consisted exclusively of patients with metastatic disease.

Overall, our findings suggest that NECs exhibit higher mutation burdens and greater immunogenic potential than NETs, supporting the utility of CGP and immune checkpoint inhibition in this subgroup.

## Supporting information

**S1 Fig. TMB value(A) and the number of mutations(B) of NETG1–3.**
(TIF)

**S2 Fig. Treatment courses of Neuroendocrine carcinoma (A).** Kaplan–Meier curves depicting overall survival. (B, C).
(TIF)

**S1 File. GEP-NENdatasheet.**
(XLSX)

## Acknowledgments

We thank Ms. Junko Yokobori and Ms. Mika Ohki for their exceptional support of this study as clinical research coordinator (CRC). I want to thank the members of Institute of Science Tokyo, including Dr Atsushi Kudo and Dr Shinji Tanaka.

## Author contributions

**Conceptualization:** Hiroaki Ono.

**Data curation:** Suguru Miyazawa.

**Formal analysis:** Suguru Miyazawa.

**Investigation:** Hironari Yamashita, Daisuke Asano, Yoshiya Ishikawa, Shuichi Watanabe, Hiroki Ueda, Satoru Aoyama, Naoya Ishibashi, Keiichi Akahoshi.

**Supervision:** Sadakatsu Ikeda, Minoru Tanabe.

**Writing – original draft:** Suguru Miyazawa.

**Writing – review & editing:** Hiroaki Ono.

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
