## [Decision Letter · Decision Letter 0]

PONE-D-24-54138Evaluation of therapeutic agent selection based on comprehensive genomic profiling in gastrointestinal pancreatic neuroendocrine neoplasmsPLOS ONE

Dear Dr. Ono,

Thank you for submitting your manuscript to PLOS ONE. After careful consideration, we feel that it has merit but does not fully meet PLOS ONE’s publication criteria as it currently stands. Therefore, we invite you to submit a revised version of the manuscript that addresses the points raised during the review process.

We look forward to receiving your revised manuscript.

Kind regards,

Benjamin Benzon, Ph.D., M.D.

Academic Editor

PLOS ONE

Journal Requirements:

**Additional Editor Comments:**

Very nice study, reviewers have made specific comments please address them. I am looking forward to your reply and revised version of the manuscript.

Reviewers' comments:

Reviewer's Responses to Questions

**Comments to the Author**

1. Is the manuscript technically sound, and do the data support the conclusions?

Reviewer #1: Yes

Reviewer #2: Partly

Reviewer #3: Partly

2. Has the statistical analysis been performed appropriately and rigorously? 

Reviewer #1: Yes

Reviewer #2: Yes

Reviewer #3: I Don't Know

3. Have the authors made all data underlying the findings in their manuscript fully available?

Reviewer #1: Yes

Reviewer #2: Yes

Reviewer #3: Yes

4. Is the manuscript presented in an intelligible fashion and written in standard English?

Reviewer #1: Yes

Reviewer #2: No

Reviewer #3: Yes

5. Review Comments to the Author

***Reviewer #1:*** Thank you for an interesting manuscript.

Comments

1. Which mutations did you investigate?

2. How does your examination distinguish from more well-known examinations such as e.g. NGS/ whole genomic sequencing?

3. Table 1 and 2. Were there one or two patients with MEN-1? You write one in Table 1 and two in Table 2

4. Why are there only 50 patients in Table 2 and not 51 (38+12)?

5. You write that mutations in MEN1 and DAXX were found in NET patients while mutatiions in RB1 and KRAS were found in NEC. Is that a new finding or isn't it expected based on the new criteria?

6. Why are the case reports included in the paper? Should they be removed and submitted individually?

***Reviewer #2:*** It is explained in the summary that the paper deals with comprehensive genomic profiling (CGP) and that the study aimed to conduct a detailed analysis of genetic mutations and a comparison of NETs and NECs, based on the results of genetic panel testing. Additionally, this study sought to examine the frequency with which cancer gene panel test results lead to actual treatment and to evaluate treatment outcomes.

However, after reading the paper, the conclusion is that the paper should be extensively rewritten.

One of the first objections is the low number of test subjects (especially in the context of examining treatment outcomes where only 4 subjects were observed).

Secondly, a detailed analysis of genetic mutations was not conducted (e.g., combined/multiple mutations per case were not mentioned). Full names of abbreviations such as NET, NEC, and MiNEN are missing, which makes it almost impossible to follow the results without additional internet search.

English language could be improved!

The figures need to be sharpened (or possibly even changed). Additionally, the paragraphs are not structured properly—some seem randomly inserted, such as "Definition of actionability." The text should be written in a more cohesive manner.

Changes to the text:

4 - Comprehensive genomic profiling (CGP) USED to

28 – missing the explanation of abbreviations NETs and NEC

30,31 – Where were the tests conducted?

34 – What are NET G1-G3, NEC and MiNEN? Based on some paragraphs below, it can be concluded that it is WHO classification. However, to be able to follow the text, these terms must be explained better in the introduction.

53, 54 – You claim that “the incident rate of neuroendocrine tumors is relatively low”. Compared to what? What is the world statistics? What types of neuroendocrine tumours are there if gastroenteropancreatic neuroendocrine neoplasms account for “50-60% of all neuroendocrine tumors”?

67 – Replace “utilizing” with “utilisation”

95 – “sequencing at our institution” What institution?

94-97 – Rewrite the paragraph.

106 – Name the manufacturer of all kits used

117 – “Definition of actionability” Should this be a separate paragraph of incorporated in the text?

141, 142 – Rewrite as: “Based on the WHO 2019 classification, there were 5 cases of NET-G1, 7 cases of NET-G2…” It is hard to follow when using the word “respectively”

144-146 – The same comment as for row 141.

150 – “Table. 1” change to “Table 1.”

Table 1 – “Ki-67 index, median (range)” value is missing

162 – Use “comparison” instead of “comparing”.

188-194 – Make sure to write human gene names according to the correct nomenclature (big letters, italics).

199 – delete parentheses after NENs

225 – put capital C after Figure 4.

226 – parentheses are missing

233 – Nowhere in the introduction of the summary was there a mention that two case reports are a part of the paper.

References should be standardised according to the Journal guidelines. All facts throughout the text must be properly referenced!

***Reviewer #3:*** This article aims to provide a comprehensive insight into the evaluation of therapeutic agents for gastrointestinal pancreatic neuroendocrine neoplasms based on genomic profiling. While the topic is indeed interesting and holds potential for publication, several aspects require significant improvement:

1. Please ensure that all abbreviations are fully spelled out upon their first mention in the manuscript. This will enhance clarity, particularly for readers who may not be familiar with these terms.

2. Many of the abbreviations used (e.g., MiNEN, GEP-NENs, NEC, NET, NEN) are complex and quite similar, which may lead to confusion. Consider revising the choice of abbreviations to improve readability and comprehension.

3. Titles and subtitles consisting solely of abbreviations, such as the one in line 104, are uncommon. I suggest reconsidering this approach by either spelling out the abbreviation or providing a more descriptive heading to enhance readability.

4. The text would benefit from significant revision to improve fluidity and overall readability. The frequent use of abbreviations, particularly those that are similar in form, makes comprehension more challenging. Reducing their frequency or restructuring certain sections could help create a smoother reading experience.

5. The methodology section requires further elaboration. Please provide more details regarding the processing of tissue and biopsy samples to ensure methodological transparency.

6. The role of the Molecular Tumor Board should be further clarified. How were opinions aligned, and what was the decision-making process regarding treatment options? A more detailed explanation of their involvement would strengthen the manuscript.

By addressing these points, the paper’s clarity, readability, and scientific rigor will be significantly enhanced.

6. PLOS authors have the option to publish the peer review history of their article (what does this mean? ). If published, this will include your full peer review and any attached files.

**Do you want your identity to be public for this peer review?** For information about this choice, including consent withdrawal, please see our Privacy Policy .

Reviewer #1: No

Reviewer #2: No

Reviewer #3: No

---

## [Author Response · Author response to Decision Letter 1]

5 May 2025

Academic Editor

Dr. Benjamin Benzon, MD, PhD

PLOS One

Dear Dr. Benzon,

We would like to sincerely thank all the reviewers for their careful reading of our manuscript and for providing detailed and insightful comments. We have revised the manuscript substantially in response, incorporating new data and clarifying key points as suggested. Below, we provide point-by-point responses to each comment.

Reviewer #1

1. Which mutations did you investigate?

We appreciate your insightful question and apologize for the insufficient description regarding the types of mutations. In our study, we investigated mutations based on comprehensive genomic profiling using two commercially available assays: the FoundationOne® CDx panel, which covers 324 cancer-related genes, and the Guardant360® panel, which targets 74 cancer-related genes. Based on the results from these analyses, we investigated the mutations of 50 cases of neuroendocrine neoplasms (NENs), including 12 cases of neuroendocrine carcinoma (NECs). We have added the details of mutations in the Materials and Methods section.

2. How does your examination distinguish from more well-known examinations such as e.g. NGS/ whole genomic sequencing?

Thank you for raising this thoughtful point. Unlike whole genome sequencing, this study utilized two commercially available assays—FoundationOne® and Guardant360®—to perform comprehensive analysis of gene mutations using panels targeting representative cancer-related genes.

3. Table 1 and 2. Were there one or two patients with MEN-1? You write one in Table 1 and two in Table 2

We greatly appreciate your attention to detail. We apologize for the careless mistake of the description in Table 2. The correct number of patients with MEN-1 is one, as shown in Table 1. We have corrected the discrepancy in the revised manuscript.

4. Why are there only 50 patients in Table 2 and not 51 (38+12)?

Thank you for this important observation. One MiNEN case was initially included in the broader group but was later excluded to avoid confusion, as it was not considered in the final analysis focused on NETs (n=38) and NECs (n=12). The final number of cases analyzed is therefore 50.

5. You write that mutations in MEN1 and DAXX were found in NET patients while mutatiions in RB1 and KRAS were found in NEC. Is that a new finding or isn't it expected based on the new criteria?

We appreciate your thoughtful query. These findings are not novel; rather, our data support previously reported patterns, as outlined in Reference 35.

6. Why are the case reports included in the paper? Should they be removed and submitted individually?

Thank you very much for your valuable comments.　In the present study, we demonstrated the efficacy of pembrolizumab in neuroendocrine carcinomas, malignancies characterized by limited therapeutic options and high clinical aggressiveness. We believe that providing objective data is essential to support this finding. In this context, it would be preferable to include imaging data both before and after treatment. However, the current manuscript includes only radiological images without additional supporting information, which we acknowledge may be insufficient for a case report. To clarify that it is not a case report, we have revised the manuscript using different language.

What we wish to emphasize is that a relatively high proportion (over 30%) of NEC cases may exhibit a high tumor mutational burden (TMB-high). This finding highlights the potential clinical utility of comprehensive genomic profiling (CGP) in identifying NEC patients who are more likely to benefit from immune checkpoint inhibitors.

Reviewer �2

One of the first objections is the low number of test subjects (especially in the context of examining treatment outcomes where only 4 subjects were observed).

Neuroendocrine carcinoma is a rare clinical condition, and this study was conducted at a single institution. It might be true that this is a limitation of this study, and that it will be necessary to verify it at multiple institutions in the future. Despite the relatively small number of cases, we would also like to emphasize the fact that this study, which showed a high response efficacy rate for Pembrolizumab in patients with NEC, is of note.

Secondly, a detailed analysis of genetic mutations was not conducted (e.g., combined/multiple mutations per case were not mentioned).

We appreciate your insightful comment. As you suggested, we have revised the figures to include detailed information on the genetic mutations identified in each case. The updated figures now allow readers to refer to combined or multiple mutations per case.

Figure 2, 3, Table 3 and corresponding Figure legends have been added in the revised manuscript, and the Results have been changed appropriately. In the original version, only representative mutations were presented; however, we have revised Figures 2 and 3 to include more detailed mutational information. Furthermore, we have added a new Table 3 that summarizes the cases harboring actionable mutations. Additionally, data on overall survival and treatment courses of NEC cases have been included in the supplementary materials.

Full names of abbreviations such as NET, NEC, and MiNEN are missing, which makes it almost impossible to follow the results without additional internet search.

Thank you very much for your important comment. To improve clarity and readability, we have excluded the MiNEN case from the analysis, as it was complex and not relevant to the final evaluation. As a result, the study now focuses solely on pure neuroendocrine neoplasms (NENs). Additionally, we have limited the use of abbreviations to three terms—NETs (neuroendocrine tumors), NECs (neuroendocrine carcinomas), and NENs (neuroendocrine neoplasms)—and ensured that each is fully spelled out upon its first appearance in the manuscript.

The figures need to be sharpened (or possibly even changed). Additionally, the paragraphs are not structured properly—some seem randomly inserted, such as "Definition of actionability." The text should be written in a more cohesive manner.

Thank you very much for your helpful suggestion. We have revised the figures to improve their clarity and visual presentation. In particular, Figures 2 and 3 have been updated to allow for easier interpretation of the relationships among multiple gene alterations, thereby enhancing their informative value for the readers.

Changes to the text:

4 - Comprehensive genomic profiling (CGP) USED to

28 – missing the explanation of abbreviations NETs and NEC

30,31 – Where were the tests conducted?

34 – What are NET G1-G3, NEC and MiNEN? Based on some paragraphs below, it can be concluded that it is WHO classification. However, to be able to follow the text, these terms must be explained better in the introduction.

53, 54 – You claim that “the incident rate of neuroendocrine tumors is relatively low”. Compared to what? What is the world statistics? What types of neuroendocrine tumours are there if gastroenteropancreatic neuroendocrine neoplasms account for “50-60% of all neuroendocrine tumors”?

67 – Replace “utilizing” with “utilisation”

95 – “sequencing at our institution” What institution?

94-97 – Rewrite the paragraph.

106 – Name the manufacturer of all kits used

117 – “Definition of actionability” Should this be a separate paragraph of incorporated in the text?

141, 142 – Rewrite as: “Based on the WHO 2019 classification, there were 5 cases of NET-G1, 7 cases of NET-G2…” It is hard to follow when using the word “respectively”

144-146 – The same comment as for row 141.

150 – “Table. 1” change to “Table 1.”

Table 1 – “Ki-67 index, median (range)” value is missing

162 – Use “comparison” instead of “comparing”.

188-194 – Make sure to write human gene names according to the correct nomenclature (big letters, italics).

199 – delete parentheses after NENs

225 – put capital C after Figure 4.

226 – parentheses are missing

233 – Nowhere in the introduction of the summary was there a mention that two case reports are a part of the paper.

Thank you very much for your thorough and constructive feedback. We have carefully reviewed each of the points you raised above and have addressed them individually in the revised manuscript.

Reviewer #3

1. Please ensure that all abbreviations are fully spelled out upon their first mention in the manuscript. This will enhance clarity, particularly for readers who may not be familiar with these terms.

Thank you very much for your helpful suggestion. We have ensured that all abbreviations are fully spelled out upon their first mention in the manuscript to enhance clarity for all readers.

2. Many of the abbreviations used (e.g., MiNEN, GEP-NENs, NEC, NET, NEN) are complex and quite similar, which may lead to confusion. Consider revising the choice of abbreviations to improve readability and comprehension.

Thank you very much for your thoughtful feedback. In response to your suggestion, we have revised the use of disease-related abbreviations to improve readability and reduce potential confusion. Specifically, since the single MiNEN case was excluded from the final analysis, the abbreviation "MiNEN" is no longer used in the manuscript. Additionally, we have limited the abbreviations to three main terms—NECs, NETs, and NENs—to ensure greater clarity for the readers.

3. Titles and subtitles consisting solely of abbreviations, such as the one in line 104, are uncommon. I suggest reconsidering this approach by either spelling out the abbreviation or providing a more descriptive heading to enhance readability.

Thank you very much for your helpful suggestion. In response, we have revised the title in line 131 by spelling out the abbreviation and providing a more descriptive heading to enhance clarity and improve readability.

4. The text would benefit from significant revision to improve fluidity and overall readability. The frequent use of abbreviations, particularly those that are similar in form, makes comprehension more challenging. Reducing their frequency or restructuring certain sections could help create a smoother reading experience.

Thank you very much for your constructive feedback. We carefully re-evaluated the overall structure of the manuscript, with particular attention to the Introduction and Materials and Methods sections. We also revised the clarity of the figures and tables. In addition, we made efforts to reduce the frequency of similar abbreviations and improve the flow of the text to enhance readability.

5. The methodology section requires further elaboration. Please provide more details regarding the processing of tissue and biopsy samples to ensure methodological transparency.

Thank you very much for your valuable comment. In response to your suggestion, we have added further details regarding the handling and submission of tissue and biopsy samples in the Materials and Methods section to enhance methodological transparency.

6. The role of the Molecular Tumor Board should be further clarified. How were opinions aligned, and what was the decision-making process regarding treatment options? A more detailed explanation of their involvement would strengthen the manuscript.

Thank you very much for your insightful comment. In response to your suggestion, we have added a more detailed explanation in the manuscript regarding the role of the Molecular Tumor Board. Specifically, we described the types of discussions that take place during the board meetings, how consensus is reached among multidisciplinary members, and how treatment recommendations are formulated and presented to patients.

In conclusion, we would like to thank the reviewers for their careful reading of the manuscript and helpful suggestions. We believe the manuscript is much improved.

Sincerely,

Hiroaki Ono, MD, PhD

---

## [Editor Report · Decision Letter 1]

Evaluation of therapeutic agent selection based on comprehensive genomic profiling in gastroenteropancreatic neuroendocrine neoplasms

PONE-D-24-54138R1

Dear Dr. Ono,

We’re pleased to inform you that your manuscript has been judged scientifically suitable for publication and will be formally accepted for publication once it meets all outstanding technical requirements.

Kind regards,

Benjamin Benzon, Ph.D., M.D.

Academic Editor

PLOS ONE
---

## [Editor Report · Acceptance letter]

PONE-D-24-54138R1

PLOS ONE

Dear Dr. Ono,

I'm pleased to inform you that your manuscript has been deemed suitable for publication in PLOS ONE. Congratulations! Your manuscript is now being handed over to our production team.

Kind regards,

on behalf of

Dr. Benjamin Benzon

Academic Editor

PLOS ONE